# Hypothalamic Hamartomas: A Narrative Review

**DOI:** 10.3390/biomedicines13020371

**Published:** 2025-02-05

**Authors:** Marian Mitrica, Aida Mihaela Manole, Mihai Toma, Octavian Mihai Sirbu, Anca Maria Sirbu, Alice Elena Munteanu

**Affiliations:** 1Clinical Neurosciences Department, University of Medicine and Pharmacy “Carol Davila” Bucharest, 050474 Bucharest, Romania; titimitrica@yahoo.com; 2Department of Neurosurgery, ‘Dr. Carol Davila’ Central Military Emergency University Hospital, 010825 Bucharest, Romania; 3Doctoral School, Faculty of Medicine, “Carol Davila” University of Medicine and Pharmacy, 050474 Bucharest, Romania; aidamihaelamanole@gmail.com; 4Department of Neurology, ‘Dr. Carol Davila’ Central Military Emergency University Hospital, 010825 Bucharest, Romania; 5Department of Medical-Surgical and Prophylactical Disciplines, Faculty of Medicine, “Titu Maiorescu” University, 031593 Bucharest, Romania; dralicepopescu@yahoo.com; 6National Institute of Medical Expertise and Recovery of Work Capacity, Panduri 22, 050659 Bucharest, Romania; sirbu_anca_maria@yahoo.com

**Keywords:** etiology, neuromodulation, hypothalamic hamartomas, diagnostic advance, therapeutic intervention, cognitive and behavioral outcome

## Abstract

Hypothalamic hamartomas (HH) are infrequent, non-neoplastic malformations of the hypothalamus with heterogeneous clinical features, with symptoms including gelastic seizures, central precocious puberty, and cognitive or behavioral deficits. This narrative review synthesizes current knowledge regarding the etiology, clinical manifestations, diagnostic advances, and therapeutic approaches for HH. Genetic insights highlight the role of postzygotic mosaicism and dysregulated Sonic Hedgehog signaling in HH development, emphasizing their relevance in potential therapeutic strategies. Diagnostic modalities such as MRI, PET, and SEEG are pivotal in identifying and characterizing HHs, enabling precise treatment planning. Therapeutic interventions span pharmacological, surgical, and neuromodulatory approaches. While surgical approaches, such as transcallosal resection or stereotactic radiosurgery, can offer considerable seizure control, newer modalities, such as laser interstitial laser thermal therapy (LITT) as well as stereotactic radiofrequency thermocoagulation, prioritize minimizing both cognitive and behavioral sequelae. The use of pharmacologic management and neuromodulation provides adjuvant benefits, specifically in drug-resistant epilepsy; despite progress, limitations still remain, including variability of outcomes and not enough long-term studies. This review underscores the need for multidisciplinary care and advanced research to optimize outcomes and improve the quality of life for patients with HH.

## 1. Introduction

Hypothalamic hamartomas (HHs) are an uncommon condition with a reported prevalence of approximately 1 in 200,000 children and an incidence ranging from 1 in 50,000–100,000 to 1 in 1,000,000 [1,2]. Even if the debut is most common in the pediatric age group, there are also rare causes with adult-onset [3,4]. HHs consist of rare heterotopic tissues of normal neurons and glial cells, arranged disorganizedly from the ventral hypothalamus. The symptomatology is quite polymorphic, ranging from neurological manifestations to endocrine disorders accompanied by psychiatric manifestations [5]. The first is associated with epileptic seizures, in most cases with an increased risk of drug resistance, age-related delay, psychiatric and behavioral impairment. The second, parahypothalamic HH is often associated with endocrinopathies [6]. This review synthesizes the existing literature on long-term behavioral and hormonal outcomes of therapies for hypothalamic hamartoma. A literature research was conducted using PubMed, Google Scholar, Cochrane, and Web of Science electronic databases. A combination of medical subject headings terms and keywords was, used such as “hypothalamic hamartoma”, “ surgical outcome”, “behavioral outcomes”, “ hormonal outcomes” or “ endocrine outcomes”, “ surgical treatment” or “ Gamma Knife Radiosurgery” or “ laser interstitial thermal therapy”. Studies were selected based on several inclusion criteria: articles had to be published in English, with dates ranging from 2000 to October 2024. The selection process focused on the presence of HH as the main diagnosis, all types of research, from case reports and case series to clinical trials and meta-analyses that directly focus on hypothalamic hamartomas, particularly their therapies and outcomes. The information extracted from the included studies was qualitatively reviewed to identify any recurrent themes and possible gaps in understanding the chosen theme. The core exclusion criteria were studies focusing on unrelated conditions, other outcomes than those already specified, and unclear methodology of the research.

This narrative review is submitted with several limitations. It does not incorporate a formal quality assessment or meta-analysis of the included studies, possibly leading to selection bias. Other limitations are the lack of large-scale, long-term studies, variability in outcome reporting, and measurement tools. Finally, this review is based solely on existing studies and does not include original research. As a result, it may not fully reflect emerging trends or novel developments in the field.

While several reviews on hypothalamic hamartomas have been published in recent years, most have focused narrowly on specific aspects, such as surgical interventions or the genetic basis of the condition. This review aims to provide a comprehensive overview that synthesizes the latest findings across multiple domains, including pathophysiology, clinical presentation, diagnostic advancements, and emerging treatment approaches. Additionally, we have integrated recent developments and insights from studies published in the last 24 years, which were not covered in previous reviews. By addressing these gaps, this review seeks to offer a broader perspective and serve as a valuable resource for both clinicians and researchers.

## 2. Etiopathogeny

HHs are rare congenital malformations with diverse etiologies that contribute to their development. The underlying mechanisms can be broadly categorized into genetic factors, including mutations in specific genes, and histopathological features that define their structure and behavior. Understanding these etiologies provides critical insight into the formation and clinical manifestations of hypothalamic hamartomas, forming the foundation for targeted diagnostic and therapeutic approaches.

The genetic substrate of hypothalamic hamartomas is fully developed, and details on the postzygotic mosaicism mechanism have emerged since 2008. Nearly 50% of HH cases are found to be due to somatic variations in genes implicated in the Sonic Hedgehog (Shh) signaling pathway and its regulators, including GLI3 (a transcription factor) and PRKACA (a Shh repressor), or genes that encode primary ciliary proteins, in particular OFD1 [7]. The Shh signaling pathway serves an important role in neurogenesis and cell patterning during early hypothalamic development, with deregulation of this pathway driving the proliferation of the wild-type surrounding cells. GLI3 is crucial for the growth of multiple organ systems [8]. GLI3 plays an important role in the development of several organ systems, and a mutation at this level can cause various developmental abnormalities, one of which is the Pallister-Hall syndrome. It is a form of HH, but it is accompanied by bifid epiglottis, imperforate anus, polydactyly or syndactyly, and renal anomalies [9,10]. A more recent concept suggests that HH may arise from germline and somatic biallelic mutations in cilia, including genes coupled to the Shh route. The signaling role of Shh proteins relies on their transport through cilia during development. The importance of the genetic component is fundamental, as it can change the therapeutic direction of the HH.

A common element of the studies that have analyzed the histopathology of hypothalamic hamartomas is the presence of neuronal clusters. These vary from case to case in size, abundance, and density. This cluster has been described as grape-like, pleomorphic rather than spherical. Depending on the characteristics of each case, these were divided according to the predominant cell type, neuronal or glial, and their distribution, diffuse or nodular. The myelinated fibers are relatively rare and randomly distributed. They have very low Ki-67 suggesting a reduced proliferative rate. Also, cellular atypia and dysmorphism are absent. Some authors describe this neuronal cluster as the functional unit for epileptogenesis [11]. The molecular mechanisms underlying epileptogenesis remain unclear, although some studies suggest that the small neurons within these lesions possess an intrinsic membrane capacity to trigger depolarizations [12].

## 3. Clinical and Behavioral Insights

Hypothalamic hamartomas are frequently recognized by association with gelastic seizures (GS). Convulsions commonly begin in infancy but can start as early as neonatal or as far into adolescence. They are generally brief, fairly stereotypical, and often characterized by laughter-like vocalizations and facial contractions resembling a smile [13]. In early stages, in infancy, GS are usually without loss of consciousness, but autonomic signs such as facial hyperemia or pupillary dilation may occur [14]. Over time these seizures multiply generally, may come in bursts, and may be associated with altered consciousness. In certain cases, the vocalization may be in the form of crying, and the seizures are known as dacrystic.

In numerous cases, GS are associated with hormonal changes resulting from sudden activation of the sympathetic nervous system or dysfunction of the hypothalamic-pituitary axis, leading to the ictal release of gonadotropins, 17β-estradiol, and growth hormone [15].

An interesting aspect of this pathology is that it has been found that gelastic seizures with a significant emotional component are strictly hypothalamically localized. The localization is most likely extra hypothalamic when they have less emotional manifestations or other phenomena [16]. The clinical presentation of seizures may involve autonomic features, automatisms, epigastric sensations, a sense of familiarity, a feeling of unfamiliarity, crying episodes, and motor manifestations. Gelastic seizures are not necessarily pathognomonic, even if they occur in many cases. Seizures can take many forms, such as focal with impaired awareness, tonic, atonic, atypical absence, infantile spasm, and generalized tonic-clonic seizure.

Diagnosing them is frequently a difficult step because in many situations the scalp EEG can be falsely normal or with a localization that can lead to misdiagnosis.

As far as behavioral disorders are concerned, it is a challenge for both the patient and the patient’s family because the focus is on aggressiveness. Factors associated with an increased risk of developing aggression are male gender, presence of intellectual handicap, early onset, and numerous seizures [17]. In early childhood, behavioral manifestations are predominantly externalized, presenting as attention deficit, hyperactivity disorder, oppositional defiant disorder, aggressive behavior, and conduct disorder. However, as the onset occurs later in life, the behavioral symptoms tend to become internalized, manifesting as depression and anxiety, affective dysfunction, phobias, generalized anxiety, and obsessive-compulsive disorder [15].

Patients with HH exhibit a range of cognitive profiles, from normal to severely impaired. The progression of cognitive decline can be gradual and influenced by various factors. Mental decline is more frequent in patients with epileptic seizures, particularly affecting verbal memory and executive functions [18]. Studies have shown a link between the severity of cognitive impairment and increased seizure frequency, number of lesions, and increased resistance to treatment [19].

One controversial topic is sleep concerning HHS patients. Those with refractory epilepsy have a rather fragmented sleep, and this leads to a poorer quality of life. A study conducted in Germany found that patients with HHs experienced less slow-wave sleep compared to healthy individuals. Slow-wave sleep is crucial in memory consolidation in children with focal epilepsy [6]. Another study published by the American Epilepsy Society shows a significant decrease in REM sleep in patients with gelastic seizures, which led to an increase in the severity of seizures and an exacerbation of their number. There is also an article describing hypersomnia as the first symptom in a patient with HHS in the posterior region, which means that it is an area of continuous discovery [20].

The main focus of the endocrine dysfunction associated with HHs is central early puberty. This usually occurs solitarily, without any other associated hormonal changes. Even if the localization, the cause of this abnormality, is well understood and described, at the opposite pole is the pathophysiology, which presents several theories but without a clear consistency. One supposition is the pulsatile release of gonadotropins [21]. The onset of precocious puberty is considered meaningful if it starts before the age of eight in girls and nine in boys and can begin as early as one year of age. These symptoms generally include breast development, reduced stature, maturing sexual reproductive organs, deepening of the voice, pubic hair growth, and acne [22]. Other endocrine dysfunctions include acromegaly, hypogonadism and elevated cortisol levels, hypothyroidism, and growth hormone deficiency. It is worth mentioning that these disorders were evidenced before the surgical treatments because post-operative and other endocrine-related changes have been described, and they will be noted in another section below [6].

## 4. Evaluation Tools of HH

The most common morphological classification, used also for imaging, is the one that divides HHs into sessile and pedunculate (Figure 1).

One of the first investigations can be computed tomography (CT) but with a rather limited specificity. It may not be easy to have a good interpretation of axial images, but on volumetric, coronal, and sagittal scans, it may be easier to visualize. It appears as a small mass without contrast enhancement in the interpeduncular and suprasellar cisternal area, usually isodense to normal brain tissue. Occasionally small calcifications or cystic formations may appear [23].

Magnetic resonance imaging (MRI), with or without contrast, has proven its superiority in diagnosing hypothalamic hamartomas, becoming the gold standard (Figure 2). The MRI availability has led to a significant increase in the discovery rate, which inevitably leads to an improvement in therapeutic techniques. The sagittal plane is quite important because it helps to differentiate morphologic subtypes. It does not cross the blood-brain barrier, so it does not capture contrast [24].

Also, an important aspect is the early detection of HHs by fetal MRI. Because the lumen of the 3rd ventricle is not completely formed until gestational weeks 24–26, the diagnosis of HH until the first trimester is difficult. Another characteristic of HH is that their growth rate is directly proportional to brain growth; once maturation is reached, they become stationary [25].

Voxel-based morphometry analysis (VPM) is a commonly used tool to detect structural modification in gray volume. T1-weighted MRI detected a significant increase in white matter in the temporal and cerebellar lobes. This may be a major reason for the dissemination of seizure activity in patients with HH, which may lead to the occurrence of other types of seizures [26].

Functional imaging demonstrated depressed glucose metabolism in the ipsilateral hemisphere of the hypothalamic hamartoma (HH) and bilateral subcortical regions, cingulate gyrus, and cerebellum, as visualized by positron emission tomography (PET). The features of glucose hypometabolism may vary according to the type of seizure. In focal seizures with secondary bilateralization, there is hypometabolism on both sides of the precentral gyrus and in the ipsilateral insular lobule of the HH [27]. Another variant exposed by a different study showed that variations in glucose hypometabolism depend on the localization of the HH and the cortex involved in interictal and ictal discharges [28].

Single-photon emission computed tomography (SPECT), used to distinguish between the interictal and ictal phases, has shown marked hypoperfusion in the ipsilateral hypothalamus, mediodorsal thalamic nucleus, putamen, bilateral pontine tegmentum, and contralateral cerebellum. Nevertheless, the smaller hypothalamic hamartomas (HH) present a reduced sensitivity in detecting seizure activity. Therefore, due to the unpredictability and risk of false negative results, PET and SPECT are not highly recommended [29].

The electroencephalogram (EEG) is a valuable tool in this pathology, even though patients with gelastic seizures frequently have minimal or no modifications on the scalp EEG. Clinically, seizure progression is associated with frontal and temporal lobe semiology, characterized by EEG attenuation or interictal epileptiform changes. The progression may continue until the appearance of generalized tonic-clonic seizures, where interictal synchronous bilateral epileptiform activity is found [13]. Intracerebral stereoelectroencephalogram (SEEG) recordings are quite rarely used in this pathology, but there are complex situations in which you need to find the exact source of the seizures, especially as they can radiate and mimic multiple foci [30].

The combination of EEG and functional MRI, EEG-fMRI, can identify oxygen level-dependent changes that are connected to the interictal discharges detected on scalp EEG. This investigation is useful in determining the location of the epileptic center and epileptic networks. Triggered networks were found in the ipsilateral hypothalamus, contralateral cerebellum, and brainstem tegmentum, while inactivation occurred predominantly in the bilateral hippocampus. These shifts between activation and deactivation suggest that epileptiform graphoelements may disrupt neuronal pathways in the hippocampus, potentially impairing cognitive function [31].

## 5. Therapeutic Intervention

The anatomical classification of HH into intrahypothalamic and parahypothalamic types plays a crucial role in determining the clinical presentation and guiding the therapeutic approach. Intrahypothalamic HHs often require interventions targeting seizure control, including surgical resection or radiosurgery. In contrast, parahypothalamic HHs may be managed effectively with medical therapy for hormonal disturbances, reserving surgery for select cases. Given these treatments’ complexity and potential risks, a multidisciplinary approach involving neurosurgeons, endocrinologists, and neurologists is essential to tailor the optimal management plan for each patient.

The choice of technique depends on the patient’s condition, the size and location of the HH, and the expertise available. Future comparative studies and randomized trials are essential to define the optimal approach for specific patient profiles.

### 5.1. Surgical Intervention

Certain surgical procedures have been applied to improve the treatment of epilepsy secondary to hypothalamic hamartomas, depending on each patient’s clinical and anatomic particularities. Most of them reference the Delalande classification, detailed in Figure 3.

Surgical procedures can be divided according to their reasoning: resection or disconnection, imaging-assisted or functional; and their approach: minimally invasive or open. All procedures involve some resection except stereotactic radiofrequency disconnection. Open surgery includes several resections or disconnections, such as pterional/orbitozygomatic or transcallosal anterior interforniceal (TAIF) [32]. The microsurgical techniques include excision of the hypothalamic hamartoma while preserving the surrounding eloquent neurovascular structures. There are also less invasive techniques as described in Figure 4.

Even if endoscopy is a preferred method due to its advantages of fewer complications and shorter hospitalizations, it also has some limitations, such as the size of the 3rd ventricle. Factors that make a transventricular endoscopic approach more favorable are small lesions, unilateral attachment, and large ventricle size. An easier approach from the contralateral ventricle is in type 2 lesions with unilateral attachment.

The transcallosal interforniceal approach may be easier in cases involving younger patients, bilaterally attached hypothalamic hamartomas, or large lesions that fully or nearly invade the third ventricle. While this method offers relatively good seizure management, the high rate of complications restricts its use to purely cisternal peduncular forms.

In some clinics, gamma knife surgery (GKS) treats small lesions distal to radiosensitive structures in patients with high cognitive function. This type of surgery requires time to show a favorable effect; therefore, drug-resistant patients are not ideal candidates [33].

For patients with isolated gelastic seizures and pure intraventricular hamartomas, laser and radiofrequency thermocoagulation-based cutoff using robot-guided stereo-endoscopy has been successfully performed. Stereotactic radiosurgery and multiple stereotactic thermocoagulations are more effective than open microsurgery, offering seizure control rates of up to 70% with a complication rate of approximately 2% [34]. Works for all HH types and sizes and has a good outcome for nongelastic seizures. Complications commonly reported are early post-procedure puberty, weight gain, and pituitary dysfunction.

An emerging technique currently under development is stereotactic laser interstitial thermotherapy, which seems to be as equally good as open microsurgery but is associated with a reasonably elevated risk of complications. In this regard, improved targeted focusing of epileptogenic focus areas is being worked on.

### 5.2. Pharmacological Therapies

The clinical variability of cases with HHs demonstrates that there may be patients with predominantly endocrine involvement, without seizures, or with rare seizures that are therapeutically controlled, compared to patients with treatment-resistant seizures. Therefore, the decision to initiate drug treatment must consider the risk–benefit ratio for each patient. Evidence regarding optimal pharmacologic therapy is limited in this case. Much of the information related to this topic is about drug failure in epileptic seizures. Initiation is preferable with a specific antiepileptic (AED) for focal seizures, such as carbamazepine or lamotrigine. Some studies have shown some efficacy with zonisamide. Another characteristic is to take into account that most reports come from surgical centers from which most cases are, of course, drug-resistant. That is why there are very little data about patients’ follow-up under medication [35]. AED therapy faces significant challenges in managing HH due to the unique characteristics of gelastic seizures, which are often refractory to AEDs. This refractoriness arises from the intrinsic epileptogenicity of the hamartoma itself, a deep subcortical focus that AEDs, designed primarily for generalized or focal cortical networks, struggle to target effectively. Consequently, many patients require polytherapy to address multiple seizure types, increasing the risk of side effects and drug interactions [3]. Furthermore, as HH progresses, secondary epileptogenesis may develop in cortical regions, leading to secondary generalized seizures that AEDs can more effectively manage, highlighting the complex interplay between subcortical and cortical epileptogenic networks in HH [36].

### 5.3. Neuromodulation

Unlike pharmacotherapy, which targets the entire brain, or surgical treatment involving resecting, disconnecting, or ablating an anatomical region, neuromodulation uses electrical activity to disrupt or modulate epileptic activity. The most commonly used treatments for drug-resistant epilepsy are deep brain stimulation (DBS), vagus nerve stimulation (VNS), and responsive neurostimulation (RNS).

Studies on small numbers of patients show that intermittent stimulation of the left vagus nerve improved seizure control and autistic-like behavior [37].

Deep brain stimulation at the level of the anterior thalamic nucleus is a known target area in drug-resistant epilepsies. Still, unfortunately, in HHs it did not have the expected impact [38]. Instead, stimulation of the mammillothalamic tract (MMT) has proven successful. It was mainly chosen because it is more easily visualized with MRI and T2 sequences, and its diameter is much smaller than the anterior thalamic nucleus, which makes the procedure easier. One particularity is the direct avoidance of the mammillary body in order not to cause memory impairment [39].

## 6. Long-Term Cognitive and Behavioral Outcomes

### 6.1. Postsurgical Cognitive Function

The presence of cognitive impairment in patients with HH and refractory epilepsy can vary widely, but intelligence may show slight to moderate improvement after surgery, provided there are no postoperative complications. The factors influencing cognitive outcomes are not fully defined. Still, patients with the most severe cognitive impairment preoperatively and with a shorter duration of epilepsy appear to achieve the biggest improvements in intellectual performance. In a recent study, which involved 159 patients treated by transcallosal interforniceal technique surgery, it was shown that a total resection could lead to a decrease in the frequency of epileptic seizures, especially generalized seizures. This is an important factor in improving cognition and behavior disorders [40]. An essential thing is the number of operated cases and the center’s success rate. Another aspect to mention is that this technique can be used in cases where the availability of minimally invasive or non-invasive techniques is minimal or non-existent. Different neurosurgery centers have different seizure-free rates.

GKS has been used predominantly in cases with mesial temporal lobe epilepsy, with doubtful results due to side effects such as memory impairment, headaches, psychosis, dysphasia, and functional seizures. However, it showed promising results in HHs; a study published in 2021 showed that no reduction in intellectual abilities, including working memory, was detected, and no memory decline was observed in adult patients. In one study of 39 patients with HHs, significant gains were noted in several areas: working memory index improved by 46%, processing speed by 35%, total IQ by 24%, verbal comprehension index by 11%, perceptual organization index by 21%, verbal learning by 20%, and visual learning by 33%. Post-GKS, cognitive improvements were markedly more pronounced in patients who became seizure-free compared to those who did not [41].

Different hypotheses have been developed in the case of *SRT*, with the conclusion that early intervention improves postoperative cognitive status. The most positive outcome was in patients who reported interictal epileptiform discharges. Several studies have documented a correlation between the severity of cognitive decline and the presence of interictal epileptiform discharges [42,43]. Patients who had a severe grade of mental impairment or prolonged onset of epilepsy had less improvement following SRT [44]. In a study in which 100 patients were enrolled, 69% had an improvement in their IQ and reported remission of behavioral disorders [45].

Another procedure that gathers sympathy is LITT; in the data we have, there is an improvement in cognitive status and behavioral disorders. It has also been utilized in a single case of an anger outburst seizure patient but demonstrated no results [46]. It has been proven to offer superior cognitive outcomes in many situations versus open surgery [47].

SIRS is a method used with success to reduce seizures, especially in small hypothalamic hamartomas. Cognitive status and behavioral disorders have improved significantly. There were no adverse effects on executive functions or verbal and figural memory [48]. Some patients remained seizure-free or had only focal seizures that they could manage more easily, so their families noticed an increased quality of life [49].

SEEG-guided radiofrequency thermocoagulation may offer a minimally invasive and low-risk surgical approach with good results. It is indicated for deep, precisely demarcated, small, well-defined formations with good preservation of cognitive function. However, there are insufficient randomized trials demonstrating efficacy in behavioral disorders [50].

SLA has proven to be an effective method in treating hypothalamic hamartoma with a decreased number of seizures. It has proven to be a good tool, especially when focal seizures occur before they become lateralized [51], leading to the preservation of cognitive function.

### 6.2. Impact of Antiepileptic Medication on Cognition and Behavior

Most authors agreed that cognitive disability depends on certain factors such as increased seizure frequency, epileptiform changes on the EEG, and the degree of severity could be related to the size of the hamartoma. Antiepileptic medication can also contribute to this, as we know HH secondary epilepsy is in most cases drug-resistant. This situation requires high doses and a certain number of drugs, which can lead to increased occurrence of side effects [38]. Cognitive and behavioral activity is a powerful issue at different age stages; that is why it is necessary to be very careful when choosing an antiepileptic or changing/adding one. Age can be a favorable or unfavorable variable; it has been shown that initiating an antiepileptic before the age of 14 years can have a negative effect on cognition, with phenobarbital and topiramate topping the list [52]. Considering that in HH focal seizures are more prevalent, antiepileptic therapy should be based on this; thus the most used drugs are carbamazepine, lamotrigine, levetiracetam, and zonisamide. The studies found children treated with carbamazepine, valproic acid, or a combination of the two required close monitoring of cognition. However, no monitoring was needed for those treated with levetiracetam or lamotrigine.

Treatment in psychiatric disorders has relied largely on antiepileptics. Especially carbamazepine and valproic acid have shown their stabilizing role in both manic episodes and bipolar disorders. In patients with associated epilepsy, treatment with them prevents the recurrence of new episodes. This protection is dose-dependent, as the higher the dose, the higher the incidence of these psychiatric events [53].

Carbamazepine is an antiepileptic widely used and sometimes is the first intention. Small studies on children aged 3–17 have shown that within 1 year there is a degree of impairment of intelligence by changing the scale (the Stanford-Binet—Fifth Edition (SB5)). At low serum concentrations, it does not affect learning or memory but may have a moderate effect on attention, concentration, and visual memory. It may also cause some sedation and slow psychomotor activity [54]. A study investigating the long-term outcomes of carbamazepine in adult patients with epilepsy observed that patients exhibited a favorable response regarding alertness, agreeableness, and neuroticism. Nevertheless, it also had a negative impact on verbal memory, executive functions, reaction time, and attention [55].

It showed the efficacy of carbamazepine in patients with associated obsessive-compulsive disorder and paranoid ideation and improved phonetic performance in those treated with lamotrigine [56].

Lamotrigine is a newer-generation AED that is generally well tolerated in both adults and children. It has been found to have no negative impact on cognitive function in adults. Although resources are limited in children, studies have demonstrated similar positive effects in adults. When used in combination with another antiepileptic, it does not worsen pre-existing cognitive issues and may, in some cases, even improve them [57].

Levetiracetam is also a widely used antiepileptic with a good safety profile, with no negative effects on cognition and a slight improvement in executive memory. Anxiety and depression may accentuate these pathologies but on a rather small scale.

### 6.3. Neuromodulation and Cognitive Changes

Neuromodulatory therapies can improve cognition by reducing the number of seizures and their intensity and by minimizing the postictal states that can lead to memory impairment. They also help prevent the progression of epileptic encephalopathy, which causes global cognitive decline. By stabilizing epileptic networks via neuromodulation, DBS of the anterior thalamic nucleus has been demonstrated to ameliorate memory and attention. VNS has demonstrated positive effects on attention and mood, which may aid in cognitive impairment. Both VNS and DBS also reduce aggressive behavior and anxiety by modulating connections between the hypothalamus and the limbic system.

These therapies also come with certain limitations. VNS can accentuate depression and DBS by affecting the structures adjacent to the hypothalamus, which can lead to transient impairment of memory and cognition and therefore can lead to behavioral disorders, impulsivity, and confusion [16].

## 7. Long-Term Hormonal Outcomes

Most endocrine disorders occur after intervention and are not that common. They are represented by diabetes insipidus, growth hormone deficiency, central hypothyroidism, hypothalamic obesity syndrome (HOS), hyponatremia, hyperthermia, and poikilothermia. Their diagnostic and treatment methods follow the protocols for patients who have these disorders but in the absence of HH. Diabetes insipidus is a complication most often encountered in patients who undergo transcallosal resection, usually with onset in the first few days and of temporary character. It occurs in 15–55% of cases with no previously reported cases before surgery. Endocrine disorders occurred in approximately 25% of cases resected endoscopically and 50% resected both endoscopically and transcallosally. With the GSK method, weight gain was observed in 15% of the patients, and in the rest without any other noticeable changes [58]. In a study published in 2017, in which 34 patients with HH treated by GSK were followed for 2 years, only one had thyrotropin-stimulating hormone (TSH) deficiency after 2 years [59].

The complications that persist in the long term are increased appetite and a tendency towards obesity, occurring in 20% of patients operated on in this procedure [60]. The etiology of hyperphagia is complicated to establish. However, postoperative MRI can be performed to assess the degree of damage to the hypothalamus and to predict the risk of developing this type of dysfunction. Lesions in the posterior hypothalamus, including the dorsomedial nucleus and dorsal hypothalamic area, have been linked to this disorder [61]. A singular case was reported about growth hormone deficiency and hypogonadotropic hypogonadism in a 17-year-old patient, which was associated with deafness, short stature, and delayed puberty.

Because surgical resection techniques also carry the risk of endocrine disruption, minimally invasive techniques are being developed.

The impact of neuromodulation on hormone balance requires advanced research to establish the exact mechanisms by which it is influenced. What we do know is that DBS and VNS, by reducing hypothalamic overstimulation, lead to indirect improvements in neuroendocrine status.

## 8. Quality of Life and Social Integration

As a disease that presents many clinical and paraclinical aspects, it requires a multidisciplinary team. In underdeveloped or developing countries, this is a major obstacle that leads to poor management of these patients. It requires neurological, neuropsychological, cognitive, endocrine, and radiologic evaluation at least once a year. Because not uncommonly the size of the hamartoma makes it difficult to make a correct and concrete diagnosis, the imaging performed should be visualized by an expert neurologist or radiologist in HH. Five percent of patients may associate with Pallister-Hall syndrome, and there should be the possibility of genetic testing when this is suspected. Easy access to a therapeutic intervention can increase quality of life by improving cognition and decreasing the number of seizures. Also facilitating different types of interventions, invasive and minimally invasive, for better outcomes. This involves training medical staff with specialized courses to be more effective in managing these patients. Even if the surgery was a medical success and we have seizure control, possible postoperative complications need to be managed and monitored in the long term. These patients also need constant and long-term educational programs that contain learning support assistants, occupational therapists, and speech and language therapists within a specialist environment. The multidisciplinary team would need to provide support to the educator to be able to access mental health services when needed. Another problem is the transition of patients from child to adult, where they have to change part of the medical team. The lack of these centers can lead to a destabilization of the management and implicitly a decrease in the patient’s quality of life.

## 9. Conclusions

In conclusion, hypothalamic hamartomas (HHs) are complex conditions requiring a multidisciplinary approach for effective management due to their diverse neurological, endocrine, and behavioral manifestations. Advances in diagnostic imaging and minimally invasive procedures have notably improved seizure management, cognitive outcomes, and patients’ quality of life, although there remain challenges such as drug-resistant epilepsy and post-treatment endocrine dysfunction. Personalized treatment strategies adapted to individual needs, prompt interventions, and long-term follow-up are crucial to optimize outcomes. Continued research and collaboration across specialties remain essential to better understand HHs and refine therapeutic approaches for improved patient care. Larger, long-term studies are critical to validate findings, refine techniques, and establish evidence-based guidelines for optimal management of HH.

## Figures and Tables

**Figure 1 biomedicines-13-00371-f001:**
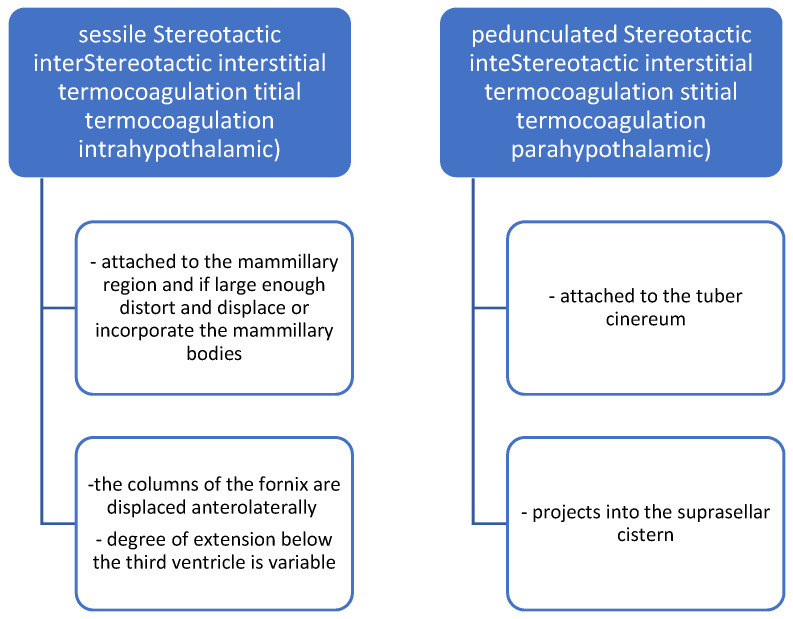
Morphologic classification.

**Figure 2 biomedicines-13-00371-f002:**
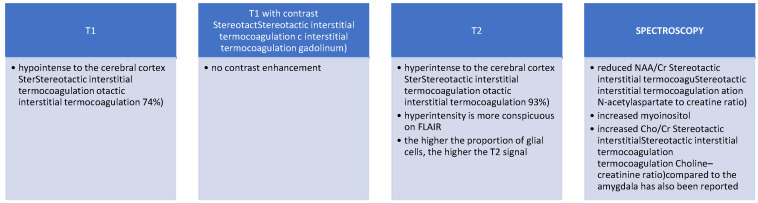
MRI findings.

**Figure 3 biomedicines-13-00371-f003:**
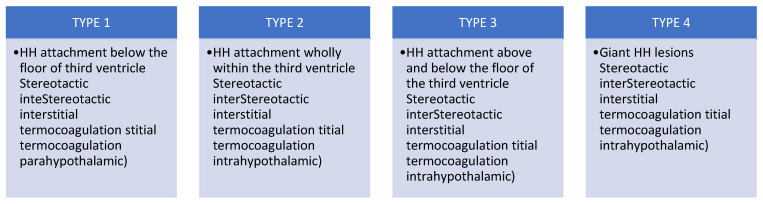
Delalande classification.

**Figure 4 biomedicines-13-00371-f004:**
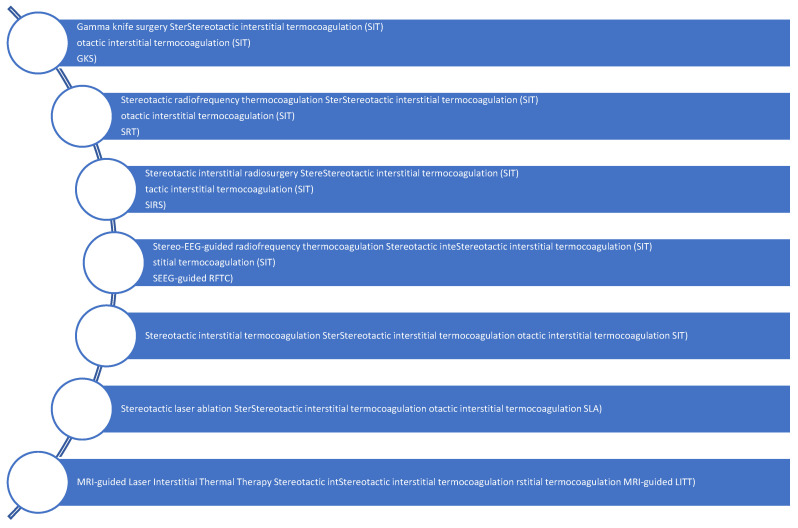
Less invasive techniques for HHs.

## Data Availability

All data are reported in the text.

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
