# Peer review of "Hypothalamic Hamartomas: A Narrative Review"

_biomedicines, 2025, doi:10.3390/biomedicines13020371_

Round 1

Reviewer 1 Report

Comments and Suggestions for Authors

This is a very comprehensive review covering various key topics from presentation, investigations, management and outcome. However, for a state-of-the-art review, the review does not seem to highlight the most recent developments or new ideas. Similarly, this review did not follow any specific format for a narrative review, systematic review, or clinical practice guidelines. I would encourage the author to select a specific methodology and follow the guidelines on how to report this. Alternatively, this can be written as a narrative review. 

1.     For the state-of-the-art review, it is advisable to follow a specific methodology. The author may refer https://pmc.ncbi.nlm.nih.gov/articles/PMC9765899/ for more information. 

2.     If a comprehensive literature search was conducted, it would be helpful to report the number of articles screened and selected in a standard format. 

3.     There are a few reviews on this topic that were published in the last few years. The author may want to explain why there is a need to perform this review and how this review is different from the previous ones. 

Reviewer 2 Report

Comments and Suggestions for Authors

In the manuscript titled Hypothalamic Hamartomas: A State-of-the-Art Review, the authors performed literature study in the latest 24 years from 2000 to Oct 2024. This review paper focused on hypothalamic hamartomas (HH) and detailed reviewed its etiology, clinical manifestations, diagnostic advances, and therapeutic approaches. However, the manuscript is disorganized and need more work to become readable as a review paper. Therefore, major revision has to be done before this manuscript being accepted for publication in Biomedicines.

Major comments: 

1. Keywords:

Since this is not a research article but a review paper, the keywords should reflect the key component of its main content. I suggest the authors considering adding “etiology”, “diagnostic advance” and “therapeutic intervention”, “cognitive and behavioral outcome”, while deleting “laser interstitial thermal therapy”, “drug-resistant epilepsy”, “transcallosal resection” and “gelastic seizure”.

2. Etiopathogeny

In this section, the authors missed a first paragraph, which should be added to summarize different etiologies which will be introduced later, such as genetics and histopathology. Readers will be lost without this summary.

In the genetic basis of HH, there are only two references in the entire paragraph, which creates doubtful credibility questionable for the readers. Same thing for the histopathology.

3. How to Handle a Clinical Chameleon

This title is too big and obscure for a review paper.

Line 133, “with externalized symptoms, figure 2, ...” There’s a need to add more words to make it coherent when reading this paragraph. I don’t see any need to add a figure here, which can be explained simply with words. Same in Figure 1 and 3. Delete these figures and add these descriptive words into the content.

4. Therapeutic intervention

Since the authors introduced two types of HH, intrahypothalamic and parahypothalamic. Will each type of HH prefer a certain type of therapeutic intervention? Please refer to more literatures and summarize this information for readers.

In the Pharmacological therapies section, there are not enough references to support your claim.

5. Figures

For all the figures, there’re no figure legends, which should be added to help reader understand. Please revise all the figures, use figure 1 as the example below.

For Figure 1, there’s no figure legend and what does the arrow mean? Dose all of these factors form a cycle?  Which is the starting point? Which is the main cause? A big circle could be used for this factor.

Minor comments: 

Some language problems existed, please consider using language polish service.

Line 44, “debut” is an informal expression, please consider using another word, such as “onset” for research/review paper.

Line 49, “trio” also sound a bit informal in a scientific paper.

Line 51, the use of punctuation. “depending on the localization : intrahypothalamic and parahypothalamic” is more accurate.

Line 122, “déjà vu, jamais vu”, what does this mean? Consider changing to other words for easy understanding.

Comments on the Quality of English Language

Some language problems existed, they may consider using a language improvement service.

Reviewer 3 Report

Comments and Suggestions for Authors

The manuscript by Mitrica et al., provides an extensive review of hypothalamic hamartoma from the point of view of its etiology, clinical manifestation, means of diagnosis, and therapy. Attention and expansion on the advance of new, non-invasive imaging approaches regarding a better diagnostic strategy-classification scheme includes state-of-the-art applications in the field: magnetic resonance imaging, fluorodeoxyglucose-positron-emission tomography, etc.-along with current diverse treatment methods for managing various symptoms by medical, neuromodulatory intervention, and/or surgery like LITT and stereo radiofrequency thermocoagulation are discussed herein. The authors have discussed with clarity the role of genetic etiologies, such as dysregulated Sonic Hedgehog signaling in HH pathogenesis, hence giving a crystal vision on the development of targeted therapies.

I have few observations to state. The review has been limited by the narrative nature with no quantitative synthesis or meta-analysis that might have strengthened the conclusion.
This paper provides important long-term cognitive, behavioral, and hormonal outcomes of interventions and again emphasizes the need for multidisciplinary care and individualized treatment plans. Although comprehensive, the review would have been more enriching if there was more detailed discussion of the newer minimally invasive techniques and their comparative efficacies. Moreover, the authors should discuss the variability in treatment outcomes and the need for long-term, large-scale studies to validate the findings presented.

With these revisions, I paper would be better for the audience of Biomedicines.

Round 2

Reviewer 1 Report

Comments and Suggestions for Authors

The author has answered the comments appropriately. However, the author has changed the format from a state-of-the-art review to a narrative review without following a standard format. This has a lower scientific value.